# Antimicrobial resistance and molecular detection of extended-spectrum ß-lactamase (ESBL)-producing *Escherichia coli* in municipal wastewater in Marrakech

**Oumaima El garraoui**[1,2], **Amal Loqman**[1,3], **Fatima-Ezahra Amouat**[1,2], **Said Hasnaoui**[1], **Nabila Soraa**[1,2], **Souad Loqman**[1,2,3]*

**1** Laboratoire de Lutte Contre les Maladies Infectieuses, Faculté de Médecine et de Pharmacie, Université Cadi Ayyad, Marrakech, Morocco, **2** Laboratoire de Microbiologie, CHU Mohammed VI, Av Ibn Sina Amerchich, Marrakech, Morocco, **3** Plateformes Biotech-Génotypage, Cité d'innovation, Université Cadi Ayyad, Marrakech, Morocco

* s.loqman@uca.ma

## Abstract

Extended-spectrum β-lactamase-producing *Escherichia coli* (ESBL-EC) poses a major threat to public health. However, its environmental dissemination, especially through wastewater, remains insufficiently addressed. This paper examines the presence of ESBL-EC in composite influent and effluent samples from Marrakech's wastewater treatment plant. Samples were screened on selective agar and isolated strains were identified by MALDI Biotyper® Sirius System. The sensitivity of bacteria to antibiotics was determined by disk diffusion method and phenotypic tests were conducted to detect ESBL production. Molecular identification was characterized by PCR and DNA sequencing. From a total of 72 wastewater samples, 364 resistant *Enterobacteriaceae* isolates were recovered, of which 134 were confirmed as ESBL-EC, with 78.35% exhibiting multidrug resistance (MDR) phenotypes. The $bla_{CTX-M}$ gene was the most prevalent among isolates obtained from influent samples, followed by $bla_{TEM}$ at 81.25% and $bla_{SHV}$ at 16.67%. Although biological treatment reduced the number of ESBL-EC isolates harboring resistance genes, a significant proportion remained detectable in the final effluent, even after UV disinfection, highlighting the limited efficacy of standard treatment processes in eliminating them. These findings underscore the urgent need to enhance wastewater treatment strategies. Implementing advanced technologies, such as membrane filtration or ozonation, combined with routine monitoring, is critical to reduce the environmental release of resistant bacteria and mitigate their public health risks.

**Data availability statement:** All relevant data are within the manuscript.

**Funding:** The author(s) received no specific funding for this work.

**Competing interests:** The authors have declared that no competing interests exist.

## Introduction

Antibiotic-resistant bacteria (ARB) have been categorized as a global health crisis and identified by the World Health Organization (WHO) as one of the top ten public health problems [1]. The One Health approach addresses this challenge by examining the interconnections of microbial populations in humans, animals, and the environment [2–5].

Wastewater treatment plants (WWTPs) play a significant role in ARB dissemination, serving as reservoirs for antibiotic resistance genes (ARGs) and facilitating their spread into surface waters and soils [6]. As WWTPs often fail to achieve complete microbial removal, ARB and ARGs may persist in treated effluents [7–9]. Investigations in Peru, Brazil, and the United States have confirmed the presence of resistant bacteria in wastewater, raising concerns about their environmental impact and potential reintroduction into clinical settings [10]. A recent study in Norway further reinforced these concerns by demonstrating that WWTPs are key reservoirs for ARB and ARGs, with numerous heterotrophic bacteria in effluents exhibiting resistance to multiple antibiotics [11]. Additionally, a systematic review concluded that antibiotic resistance in environmental samples surrounding WWTPs may pose occupational health risks for workers and potential exposure risks for nearby residents [12].

Among ARB, Gram-negative bacteria from the *Enterobacteriaceae* family are the most concerning, particularly *Escherichia coli*, which have developed a fast-paced resistance primarily through the production of extended-spectrum beta-lactamase (ESBL) enzymes [13–15]. ESBL-EC is of particular concern due to its ability to overcome commonly used antibiotics. ESBLs, primarily encoded by $bla_{CTX-M}$, $bla_{TEM}$, and $bla_{SHV}$ genes, are often carried on mobile genetic elements such as plasmids, enabling horizontal gene transfer and further dissemination of resistance [16–18]. ESBL-EC has been detected in wastewater across several countries. For instance, a survey across 12 WWTPs in Tokyo found that 5.7% of *E. coli* isolates were resistant to cefotaxime, with ESBL-EC accounting for 5.3% of the total *E. coli* population in chlorinated effluents [19]. A Swiss study reported variations in ESBL-EC concentrations, with higher levels observed in plants serving larger populations [20]. In South Africa, a study assessed 90 *E. coli* isolates that survived chlorination [21].

In Morocco, research on ESBL-EC has primarily focused on clinical settings [22–25], and has been reported in food products, including raw meat and poultry, as well as in catering services [26–28]. However, environmental studies are relatively limited, especially concerning WWTPs [29–31].This is particularly concerning given the country's rapid urbanization and growing wastewater reuse practices. This study aims to address this gap by investigating the resistance profiles and encoding genes of ESBL-EC in the influent and effluent of the Marrakech WWTP. The findings will provide the first comprehensive environmental analysis of ESBL-EC in Moroccan wastewater, offering insights into its potential risks and informing future antimicrobial resistance mitigation strategies.

## Materials and methods

### Sampling and study locations

The Marrakech wastewater treatment plant is located 13 km from Marrakech, on the national road No. 7. The wastewater treatment plant is situated northwest of the city, on the left bank of the Tensift River. It covers an area of 17 ha with a potential of 237.000 m³/day comprising community, hospital, and slaughterhouse in flow.

The WWTP employs a two-stage treatment process to reduce microbial contaminants. During the secondary or biological treatment, the influent undergoes aerobic activated sludge treatment in aerated reactor tanks, contributing to the reduction of bacterial load. Following biological treatment, the effluent undergoes a tertiary treatment that includes coagulation, flocculation, sand filtration, ultraviolet disinfection (UV), and chlorination, which targets microbial DNA to inactivate pathogenic and antibiotic-resistant bacteria. Wastewater samples were collected over one year from influent (raw wastewater) and effluent (treated water). A total of 72 composites 24-hour flow-proportional samples were collected to ensure representative results. This sample size was determined based on previous studies assessing wastewater-based antibiotic resistance and was selected to balance feasibility with statistical significance [32]. Each 1 L sample was transported on ice (4–8°C) to the laboratory and analyzed within two hours to minimize bacterial degradation.

### Characterization of the bacterial isolates

Wastewater samples were filtered using 0.45 µm sterilized filter membrane (Millipore, USA). The filtered material was inoculated on MacConkey agar (Bio-Rad Laboratories, USA) with and without 4 mg/L of ceftriaxone. Colonies showing a flat, dry, pink, non-mucoid appearance with a darker pink zone of precipitated bile salts, typical of *E. coli*, selected from MacConkey agar supplemented with 4 mg/L of ceftriaxone were further subjected to MALDI-TOF (Matrix-Assisted Laser Desorption/Ionization Time-of-Flight) Biotyper® Sirius System (Bruker, Germany) [33–35].

### Antimicrobial susceptibility testing

Antimicrobial susceptibility testing (AST) was performed on ESBL-EC isolates using the standardized disk diffusion method on Mueller-Hinton agar (Biolabs, Budapest, Hungary), following the CASSFM/EUCAST 2019 guidelines. The used antibiotic discs (Oxoid™) list is: Amikacin (AK, 30 µg), amoxicillin/clavulanic acid (AMC, 30 µg), aztreonam (ATM, 30 µg), cefepime (CEP, 30 µg), ceftriaxone (CRO, 30 µg), ceftazidime (CAZ,10 µg), cefotaxime (CTX, 5 µg), cefoxitin (FOX, 30 µg), meropenem (MEM,10 µg), ertapenem (ETP, 10 µg), fosfomycin (FF, 10 µg), gentamicin (GM,10 µg), imipenem (IMP,10 µg), ciprofloxacin (CIP,5 µg), and trimethoprim/sulfamethoxazole (SXT, 25 µg). Quality control was carried out using *E. coli* ATCC 25922, *Klebsiella pneumoniae* ATCC 180112, and *Enterobacter cloacae* ATCC 180083 as negative controls, and *E. coli* ATCC 151006, *Klebsiella pneumoniae* ATCC 180111, *Enterobacter cloacae* ATCC 161002 as positive controls. The inoculum's concentration was normalized to 0.5 McFarland turbidity, and the plates were incubated for 18–20 h at 35 ∘C and assessed for the formation of inhibition zones. The zone diameters were interpreted based on the EUCAST clinical breakpoints. AST results were further confirmed using the Phoenix™ Automated Microbiology System (Becton–Dickinson Diagnostic Systems, Sparks, MD, USA) [36].

### Phenotypic detection of ESBL production *E. coli*

Ceftriaxone-resistant *E. coli* isolates were screened for ESBL production using CHROMID® ESBL agar (Biomerieux, France) according to the manufacturer's recommendation. Plates were incubated for 18–24 h at 36 ± 2 °C. Colonies recognized as ESBL-EC were confirmed with the Double Disk Synergy Test (DDST) [37,38]. The test is based on placing each disk (Oxoid™) containing 30 µg of cefotaxime, ceftazidime, or cefepime was set close to a disk carrying amoxicillin-clavulanic acid (30 µg) on plates with Mueller-Hinton agar plates inoculated with the corresponding isolates. A

keyhole-shaped enhancement or an increase in the inhibition zone between clavulanic acid and one of the chosen cephalosporins was identified as positive for ESBL production.

### ESBL encoding genes

Bacterial DNA extraction was performed using the boiling lysis method [39]. ESBL-producing isolates were cultured on Luria Bertani agar and incubated overnight at 37 °C. A single colony was suspended in 200 μL of nuclease-free water (Invitrogen, Paisley, UK), boiled at 100 °C for 10 min in a thermal block (Polystat 5, Bioblock Scientific, Illkirch, France), and centrifuged at 19,000 × g for 5 min. The presence of resistance genes was screened using real-time PCR with specific primers targeting ESBL encoding genes ($bla_{CTX-M}$ (different groups), $bla_{TEM}$, and $bla_{SHV}$) as previously described [40,41]. A susceptible *E. coli* strain was included as a negative control to validate the specificity and accuracy of the qPCR assay, ensuring that amplification signals were only detected in ESBL-producing isolates. To ensure data reproducibility, all reactions were performed in triplicate.

Isolates that tested positive for beta-lactamase genes were subjected to conventional PCR to confirm the previous results. PCR products were purified and sequenced using the Big Dye terminator chemistry on an ABI 3130XL automated sequencer (Thermo Fisher Scientific, Waltham, MA, USA). The resulting sequences were analyzed using CodonCode Aligner software (version 3.7.1.1) and examined using the BlastN and BlastP then compared to the ARG-ANNOT (Antibiotic Resistance Gene-ANNOTation) database and NCBI GenBank database (www.ncbi.nlm.nih.gov).

### Sample storage, disposal, and ethical considerations

All wastewater samples were handled following biosafety and environmental regulations. Samples were stored at 4°C for a maximum of 24 hours before processing. While extracted DNA was preserved at −80°C for future analysis. Post-analysis wastewater and bacterial cultures were autoclaved at 121°C for 15 minutes before disposal to prevent environmental contamination. The study complied with Moroccan environmental regulations, with sampling approval obtained from the Water and Electricity Distribution Authority of Marrakech (RADEEMA-Régie Autonome de Distribution d'Eau et d'Electricité de Marrakech). As wastewater samples were collected from public treatment facilities and did not involve human subjects, no additional ethical approvals were required.

## Results

### Prevalence of resistant *E. coli* and ESBL-producing *E. coli*

A total of 72 wastewater samples were collected, including influent (n = 25), biologically treated effluent or secondary treatment (n = 24), and tertiary treatment effluent (n = 23). Of these, 69 samples (95.83%) tested positive for *E. coli*, including all treated effluent samples.

Out of 364 resistant *Enterobacteriaceae* isolates recovered from selective agar, 264 were confirmed as resistant *E. coli*, representing 72.52% of the total resistant isolates. The proportion of resistant isolates was higher in influent samples (n = 146; 55.30%) compared to biologically treated effluent (n = 75; 28.40%) and tertiary treatment effluent (n = 43; 16.28%). The data indicates that the number of resistant isolates decreased after wastewater treatment, particularly tertiary treatment, but they were not eliminated completely.

The phenotypic detection and the confirmation with selective Chrom ID ESBL resulted in the identification of 134 ESBL-EC (50.76%) among the 264 resistant *E.coli*. The display of the phenotypic characteristics of *E. coli* isolates is illustrated in Fig 1. Section (a) shows *E. coli* growth on MacConkey agar (Bio-Rad Laboratories) with 4 mg/L ceftriaxone, section (b) displays growth on CHROMID® ESBL agar, and panel (c) shows the characteristic 'champagne-cork' appearance indicative of ESBL production.

These visual confirmations support the phenotypic identification of ESBL production, aligning to accurately characterize resistant *E. coli* strains.

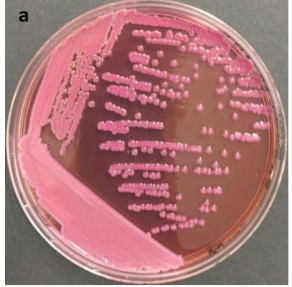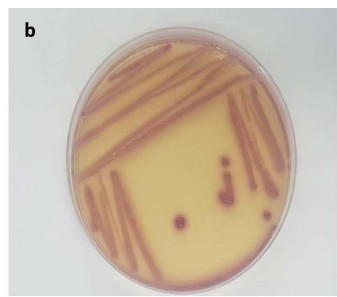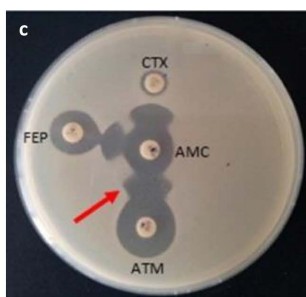

**Fig 1. _E. coli_ growth and ESBL phenotypic characteristics.** (a) MacConkey Agar with 4 mg/L ceftriaxone, (b) CHROMID® ESBL Agar, and (c) Phenotypic Test Showing 'Champagne-Cork' Appearance Indicative of ESBL Production.

## Antibiotic susceptibility testing

The ESBL-EC strains (n = 134) showed variable susceptibility to the tested antibiotics, with isolates from the influent demonstrating an increased resistance rate compared to isolates from the effluent. 50% of the isolated strains showed resistance to 3rd generation cephalosporins (cefotaxime and ceftazidime). The resistance profile indicates that 91.04% were resistant to amoxicillin-clavulanic acid, 50% of the isolates were resistant to trimethoprim/sulfamethoxazole, 32.08% to ciprofloxacin, and 27.67% to gentamicin. Resistance to meropenem was less frequently observed, with a rate of 1.56%. This indicates that resistance to commonly used antibiotics is prevalent, particularly in influent samples.

A percentage of 78.35% (n = 105) of ESBL-EC isolates illustrated multiple drug resistance (MDR) phenotypes. Even though the majority of isolates were resistant to at least two antibiotic classes, resistance beyond three antibiotics was observed. This high prevalence of MDR phenotypes highlights the selective pressure exerted by wastewater environments, directly contributing to the risk of multidrug-resistant pathogens in effluent discharge.

Among ESBL-producing isolates from treated wastewater, resistance to fluoroquinolones, aminoglycosides, and tetracycline appeared lower compared to isolates from influent. These profiles demonstrate a reduction in resistance from influent to tertiary treated effluent. An outline of the antimicrobial resistance of the analyzed strains from each treatment site is shown in Fig 2.

## ESBL-genotypes

A total of 134 ESBL-EC isolates were confirmed using phenotypic methods, of these 125 isolates underwent molecular analysis for the detection of $bla_{CTX-M}$, $bla_{TEM}$, and $bla_{SHV}$ genes. The distribution of these resistance genes was examined across the three wastewater treatment sites: influent (Site 1, n = 96), effluent after biological treatment (Site 2, n = 17), and effluent after UV disinfection (Site 3, n = 12).

The $bla_{CTX-M}$ gene was the most frequently detected in 100% of isolates across all sites. $bla_{TEM}$ was highly prevalent, with 81.25% of isolates from Site 1, 82.35% from Site 2, and 83.33% from Site 3. In contrast, $bla_{SHV}$ showed a lower frequency but was detectable at all sites. The co-occurrence of multiple genes, including $bla_{CTX-M} + bla_{TEM} + bla_{SHV}$, was observed in 12.5% of isolates in Site 1, 11.76% in Site 2, and 16.67% in Site 3.

Table 1 presents the frequency (%), calculated as the proportion of isolates positive for each gene out of the total ESBL-EC at each site. These findings highlight the persistence and complexity of ESBL genes throughout wastewater treatment, posing a public health risk due to the potential dissemination of antimicrobial resistance.

## Discussion

Wastewater treatment plants are recognized as hotspots for the dissemination of antibiotic-resistant bacteria and resistance genes in the environment [42–45]. Among these, ESBL-EC has emerged as a major threat to public health due to

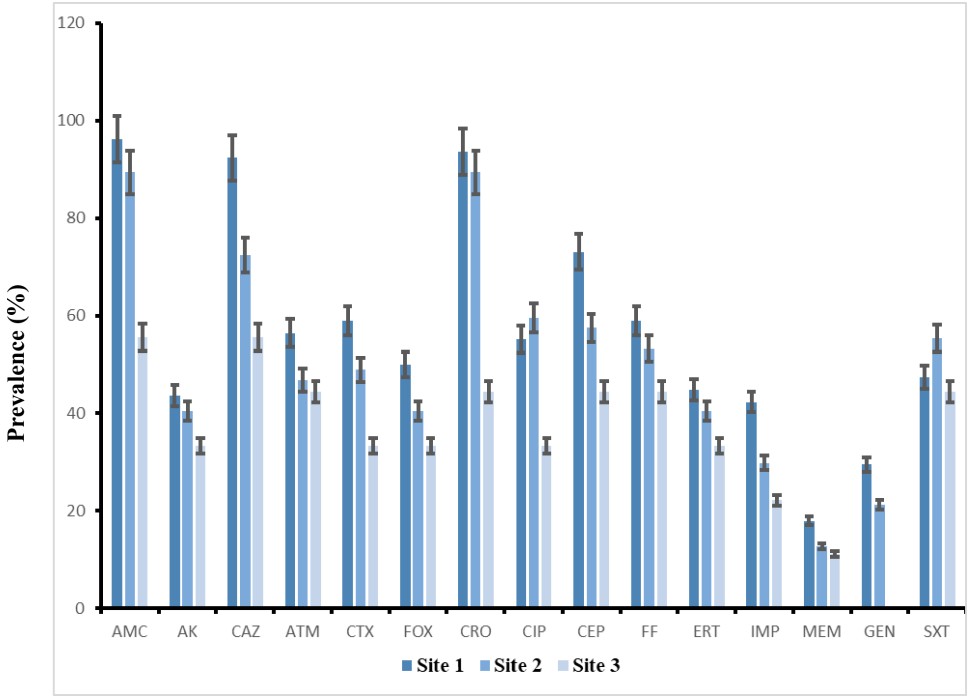

**Fig 2. Antimicrobial resistance percentages for *E. coli* strains across the three treatment sites.** Site 1: influent, Site 2: effluent after biological treatment, Site 3: effluent after the UV and chlorination treatment. Antibiotic (amoxicillin clavulanic acid (AMC), amikacin (AK), ceftazidime (CAZ), aztreonam (ATM), cefotaxime (CTX), cefoxitin (FOX), ceftriaxone (CRO), ciprofloxacin (CIP), cefepime (CEP), fosfomycin (FF), ertapenem (ERT), imipenem (IMP), meropenem (MEM), gentamicin (GEN), trimethoprim/sulfamethoxazole (SXT)).

**Table 1. Genotypes of ESBL-producing *Escherichia coli* resistance genes detected in wastewater from three different sites (n = 125).**

|  | SITE 1[a] |  | SITE 2[b] |  | SITE 3[c] |  |
|---|---|---|---|---|---|---|
| **Total ESBL EC** | **96** |  | **17** |  | **12** |  |
| **Resistant Genes** | **n** | **Frequency %** | **n** | **Frequency %** | **n** | **Frequency %** |
| ***blaCTX-M*** | 96 | 100 | 17 | 100 | 12 | 100 |
| $bla_{CTX-M-1}$ | 35 | 36.46 | 9 | 52.94 | 5 | 41.67 |
| $bla_{CTX-M-15}$ | 32 | 33.33 | 4 | 23.53 | 3 | 25 |
| ***blaTEM*** | 78 | 81.25 | 14 | 82.35 | 10 | 83.33 |
| ***blaSHV*** | 16 | 16.67 | 3 | 17.65 | 2 | 16.67 |
| ***blaCTX-M+blaTEM+blaSHV*** | 12 | 12.5 | 2 | 11.76 | 2 | 16.67 |

n: number of isolates for each site,

[a]: Site 1: influent,

[b]: Site 2: effluent after biological treatment,

[c]: Site 3: effluent after UV treatment and chlorination. Total ESBL EC: Total number of ESBL-producing *E. coli* isolated at each site. Frequency %: Proportion of ESBL-EC isolates carrying each resistance gene at each site.

its increasing prevalence and persistence in wastewater systems. This study provides an extended screening of ESBL-EC isolates in both influent and effluent samples at different stages of treatment in the WWTP of Marrakech.

Out of 72 collected samples, 69 (95.83%) tested positive for the presence of resistant *E.coli*, including all treated effluent samples. These results align with the study by Nzima et al., which showed that WWTPs serve as major sources

of multidrug-resistant *E.coli*, presenting an emerging environmental and public health risk. [46]. The highest prevalence of resistant strains was observed in influent samples, likely due to fecal contamination from various sources such as hospitals, households, and agricultural runoff [47,48]. In addition, seasonal variations may influence the prevalence of resistant strains, as fluctuating temperatures and varying wastewater inflows could affect bacterial survival and gene persistence [49].

Although chlorine disinfection and UV treatment decrease the number of resistant *E. coli* isolates, our findings along with previous studies indicate that resistant isolates remain present in the treated effluent [50–54]. This persistence suggests that conventional treatment processes are insufficient in degrading ARGs, allowing their release into the environment. Horizontal gene transfer mechanisms, such as plasmid-mediated exchange, contribute to this persistence, enabling resistance genes to survive under difficult conditions. Furthermore, certain resistance genes may be present in extracellular DNA fragments, which are not effectively removed by standard disinfection methods [55]. This limitation necessitates the adoption of advanced treatment technologies. Potential alternatives, such as membrane bioreactors, ozonation, activated carbon adsorption, and advanced oxidation processes, should be explored for their efficiency in degrading resistance genes in wastewater.

Antibiotic susceptibility testing indicated widespread resistance to amoxicillin-clavulanic acid, while intermediate resistance was observed to third-generation cephalosporins (cefotaxime and ceftazidime), consistent with prior reports from Ferreira da Silva et al [56]. However, the relatively low resistance to gentamicin shows that aminoglycosides may still be effective against certain strains in wastewater [57,58].

Molecular analysis confirmed that 100% of ESBL-EC isolates carried $bla_{CTX-M}$ across all sites. Among the $bla_{CTX-M}$ subgroups, $bla_{CTX-M-1}$ and $bla_{CTX-M-15}$ were the most prevalent, ranging from 36.46% to 52.94% across different sites. This observation is consistent with the finding of Hassen et al., who reported a predominance of bla genes, with 28 of the isolates producing $bla_{CTX-M-15}$ [59]. Similarly, a study by Lidhegner et al showed that the ESBL description of cefotaxime-resistant populations recognized $bla_{CTX-M}$ subgroup as prevalent [60]. Their prevalence was in terms with many studies conducted in France [61], and Portugal [50], as well as the finding of B. Bojar et al., that emphasizes a high incidence of $bla_{CTX-M-1}$ [62]. The presence of mobile genetic elements, such as plasmids encoding *bla* genes, increases the risk of resistance gene exchange among bacterial populations, which may lead to the emergence of more resistant strains over time. These findings suggest that $bla_{CTX-M}$ genes may serve as key molecular markers for monitoring resistance dissemination in wastewater systems [63]. A concerning finding is the coexistence of $bla_{CTX-M}$ with other β-lactamase genes, such as $bla_{TEM}$, which suggests the potential for multi-drug resistance (MDR) accumulation through genetic recombination [64,65].

This research has illustrated the incidence of ESBL-EC in wastewater from the WWTP in Marrakech, confirming the plant's role as a hotspot for the dissemination of antibiotic-resistance genes. Their prevalence within the research region urges the need to enhance decontamination strategies, especially for wastewater from various sources, such as hospitals. Furthermore, the high detection rate of $bla_{CTX-M}$ genes implies their possible application as biomarkers for monitoring resistance dissemination and guiding targeted interventions to reduce public health risks, supporting the One Health approach, linking environmental, human, and animal health through the tracking of ARGs in wastewater. Further research is necessary to assess the diversity of ESBL-producing *Enterobacteriaceae* in the environment, determine *E. coli* phylogenetic groups, and evaluate novel treatment approaches to limit the persistence of resistant bacteria in wastewater environments.

## Conclusion

This research work was undertaken at the WWTP of Marrakech to determine the resistance profiles of ESBL-EC using both phenotypic and molecular techniques. Our findings indicate that the effluent of WWTPs is contaminated with ESBL-EC, which significantly contributes to the emergence of antibiotic-resistant bacteria and the spread of their resistance genes. Such contaminants discharged into the environment may result in potential health impacts on humans and

animals. To mitigate these risks, policymakers must reinforce wastewater treatment regulations and implement stricter effluent standards. Interdisciplinary collaborations between microbiologists, environmental engineers, and public health officials are essential for developing integrated strategies to combat antibiotic resistance. The adoption of advanced treatment technologies, including advanced oxidation processes, is essential to enhance bacterial removal and gene degradation. Further studies are needed to investigate their spread into the environment and identify possible pathways of dissemination.

## Acknowledgments

The authors gratefully acknowledge the National Office of Electricity and Drinking Water as well as the Autonomous Water and Electricity Distribution Authority in Marrakech City for allowing us to collect the wastewater samples.

## Author contributions

**Conceptualization:** Souad Loqman.

**Data curation:** Oumaima El garraoui, Amal Loqman, Fatima-Ezahra Amouat.

**Funding acquisition:** Souad Loqman.

**Methodology:** Oumaima El garraoui, Amal Loqman, Fatima-Ezahra Amouat, Said Hasnaoui, Souad Loqman.

**Project administration:** Souad Loqman.

**Software:** Oumaima El garraoui, Amal Loqman, Fatima-Ezahra Amouat, Said Hasnaoui.

**Supervision:** Souad Loqman.

**Writing – original draft:** Oumaima El garraoui.

**Writing – review & editing:** Nabila Soraa, Souad Loqman.

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
