## [Decision Letter · Decision Letter 0]

7 Feb 2025

*Escherichia coli*

Dear Dr. Loqman,

Thank you for submitting your manuscript to PLOS ONE. After careful consideration, we feel that it has merit but does not fully meet PLOS ONE’s publication criteria as it currently stands. Therefore, we invite you to submit a revised version of the manuscript that addresses the points raised during the review process.

We look forward to receiving your revised manuscript.

Kind regards,

Gabriel Trueba, PhD

Academic Editor

PLOS ONE

Journal requirements:   When submitting your revision, we need you to address these additional requirements. 1. Please ensure that your manuscript meets PLOS ONE's style requirements, including those for file naming. The PLOS ONE style templates can be found at https://journals.plos.org/plosone/s/file?id=wjVg/PLOSOne_formatting_sample_main_body.pdf and https://journals.plos.org/plosone/s/file?id=ba62/PLOSOne_formatting_sample_title_authors_affiliations.pdf. 2. Please amend either the title on the online submission form (via Edit Submission) or the title in the manuscript so that they are identical. 3. We note that your Data Availability Statement is currently as follows: [All relevant data are within the manuscript and its Supporting Information files.] Please confirm at this time whether or not your submission contains all raw data required to replicate the results of your study. Authors must share the “minimal data set” for their submission. PLOS defines the minimal data set to consist of the data required to replicate all study findings reported in the article, as well as related metadata and methods (https://journals.plos.org/plosone/s/data-availability#loc-minimal-data-set-definition). For example, authors should submit the following data: - The values behind the means, standard deviations and other measures reported;- The values used to build graphs;- The points extracted from images for analysis. Authors do not need to submit their entire data set if only a portion of the data was used in the reported study. If your submission does not contain these data, please either upload them as Supporting Information files or deposit them to a stable, public repository and provide us with the relevant URLs, DOIs, or accession numbers. For a list of recommended repositories, please see https://journals.plos.org/plosone/s/recommended-repositories. If there are ethical or legal restrictions on sharing a de-identified data set, please explain them in detail (e.g., data contain potentially sensitive information, data are owned by a third-party organization, etc.) and who has imposed them (e.g., an ethics committee). Please also provide contact information for a data access committee, ethics committee, or other institutional body to which data requests may be sent. If data are owned by a third party, please indicate how others may request data access.

Reviewers' comments:

Reviewer's Responses to Questions

**Comments to the Author**

1. Is the manuscript technically sound, and do the data support the conclusions?

Reviewer #1: Partly

Reviewer #2: Partly

2. Has the statistical analysis been performed appropriately and rigorously?

Reviewer #1: Yes

Reviewer #2: No

3. Have the authors made all data underlying the findings in their manuscript fully available?

Reviewer #1: Yes

Reviewer #2: No

4. Is the manuscript presented in an intelligible fashion and written in standard English?

Reviewer #1: Yes

Reviewer #2: No

Reviewer #1: Manuscript Review: "Antimicrobial Resistance and Molecular Detection of Extended-Spectrum β-Lactamase (ESBL)-Producing Escherichia coli in Municipal Wastewater in Marrakech"

The title should read "Antimicrobial Resistance and Molecular Detection of Extended-Spectrum β-Lactamase (ESBL)-Producing Escherichia coli in Municipal Wastewater in Marrakech: Implications for Public Health and Environmental Contamination."

The abstract should be improved by incoporating the following:

Revise sentences to avoid redundancy and enhance readability. For instance, the phrase: “emphasizing the persistence of these resistant genes even after disinfection processes” could be streamlined to: “highlighting the limited efficacy of disinfection in eliminating resistant genes.”

Emphasize practical applications, e.g., how findings can inform wastewater treatment protocols.

Avoid excessive numerical data in the abstract; instead, provide a general overview of key findings.

The introduction could be improved as follows:

Enhance the clarity of global relevance by briefly discussing similar findings in other regions.

The literature review could benefit from integrating more recent studies (2023–2024) to reflect cutting-edge research on wastewater surveillance.

State the research gap more explicitly, emphasizing why Morocco’s wastewater systems need targeted investigation.

The methods could be improved as follows:

The rationale for selecting 72 samples could be expanded to address whether this number is statistically sufficient to generalize findings.

While the description of tools and techniques (e.g., MALDI Biotyper, RT-PCR) is clear, specific details on reagent sources and controls need clarification to enhance reproducibility.

Although the study involves wastewater samples, a brief mention of ethical considerations or permits for sample collection would strengthen compliance reporting.

Include a section discussing data reliability and replication, especially for RT-PCR and sequencing steps.

Clarify the storage or disposal methods for wastewater samples post-analysis.

The results could be improved as follows:

Tables 1 and Figures 1 and 2 contain excessive detailed data, which might be overwhelming for readers.

Results focus heavily on resistance gene profiles but lack an in-depth interpretation of patterns and trends.

Simplify tables by consolidating similar categories and highlighting the most relevant data.

Include additional visual aids, such as heatmaps, to better illustrate resistance patterns across sample sites.

Provide a brief summary for each result to connect data points to the research objectives.

The discussion could be improved as follows:

The section occasionally reiterates results instead of critically analyzing them.

Lack of detailed suggestions on how findings could influence wastewater treatment improvements.

Does not address the limitations of the study, e.g., the exclusion of seasonal variations or genetic typing limitations.

Critically evaluate the persistence of genes post-disinfection and propose possible mechanisms.

Highlight practical interventions for improving wastewater treatment protocols based on findings.

Discuss broader implications for global health surveillance, aligning with the One Health approach.

Explicitly state study limitations and suggest future research directions, such as assessing resistance dynamics across seasons or expanding to other wastewater sites.

The conclusion could be improved as follows:

Add a call to action for policymakers to tighten water treatment regulations.

Suggest the adoption of advanced treatment technologies or routine monitoring programs.

Highlight the need for interdisciplinary collaborations to address antibiotic resistance in wastewater.

The Figures and tables could be improved as follows:

Ensure figures are labeled with legends that clearly explain all components.

Add captions that emphasize key takeaways for each table or figure.

Reduce repetitive information to maintain reader engagement.

The references could be improved as follows:

Double-check for citation consistency, particularly with formatting styles.

Include a few more references from 2023–2024 to strengthen the manuscript’s currency. Examples include

https://doi.org/10.1038/s41598-024-72993-w

https://doi.org/10.1128/mra.00140-24

https://doi.org/10.1093/sumbio/qvae017

https://doi.org/10.1016/j.lwt.2023.114913

The manuscript grammar and sentence structures could be improved as follows

Revise long sentences for clarity and avoid excessive technical jargon where simpler language can suffice.

Conduct thorough proofreading to eliminate typographical errors.

Reviewer #2: Thank you for sending me the manuscript by Garraoui et al. This manuscript reported the antimicrobial resistance and molecular characteristics of ESBL-producing E. coli in municipal wastewater in Marrakech. In general, the study is technically sound. However, several issues related to data presentation and interpretation should be addressed. Particularly, the data presented in Table 1 and the related interpretation should be revised to draw correct conclusions.

Materials and Methods section:

- Ln 92: please describe the effluent treatment method. In addition, the Results section mentioned about 2 stages of treatment: biological treatment and UV treatment, so please describe the details in this section.

- Ln101: please describe the “E. coli-like appearance” and cite the reference(s).

- Ln 102: please add the purpose of the MALDI method.

- Ln 105: antimicrobial susceptibility test was performed on which strains, and on how many strains in total?

- Ln 124-129: please add reference(s) for the Double Disk Synergy Test.

- Ln131: please note that “RT-PCR” stands for “Reverse transcription-PCR”, not “Real-time PCR”. Which method did the authors perform? Please also describe the PCR protocol after the boiling lysis step in this paragraph.

- Ln 134: why did the authors choose the “susceptible” E. coli strain? How about the resistant strains?

- Ln 139: what does “positive isolates” mean?

- Ln 139-142: the authors mentioned about DNA sequencing and BlastN and BlastP analysis, but no data of these analyses were reported in the Results section!

Results section:

- Ln 149: please add the percentage (%) of resistant E. coli out of all resistant strains.

- Ln 150,151: typo mistakes, please add a word space before and after “;”

- Ln 155: please add the percentage (%) of ESBL-EC out of the resistant EC.

- Ln 155-159: was there any strain that was positive with Chrom ID ESBL but showed negative result with the Double Disk Synergy Test?

- Ln 168-169: “Fifty percent of the isolated strains showed resistance to 3rd generation cephalosporins”, since the authors performed several steps with different methods in this study, please clarify the method and the number of “isolated strains” in this context.

- Ln 171: grammar error: “50% trimethoprim/sulfamethoxazole”

- Ln 172: grammar error: “meropenem at 1.56 % were less frequently observed”

- Ln 173: “78.40% (n = 98)” means that 100% were 125, but no population of 125 strains appeared from the previous results. So, from which population was 78.40% calculated? Meanwhile, the number written in the Abstract was 78.35% (n = 105). Please make consistency.

- Ln 175: “was covered”: did you mean “was discovered”?

- Ln 187 and Table 1: the number of ESBL-producing E. coli was 134 but data in Table 1 showed n = 125. Please make a consistency.

- Ln 187,188: data from Table 1 did not support this statement!

- Ln 188-191: these statements should be carefully revised after revising Table 1

- Table 1: this table should be seriously revised and presented in another format considering the following flaws:

o Please provide the number of isolates from each site (site 1,2,3).

Suppose that the data of blaCTX-M is correct, considering the equality of the total number of blaCTX-M in the three sites (96 + 17 + 12 = 125) and the total isolates (n =125), the number of isolates from site 2 should not excess 17 and the number of isolates from site 3 should not excess 12. However, in the blaTEM row, the respective numbers from site 2 and site 3 were 31 and 15. Similarly, the rows of blaCTX-M+TEM+SHV or blaCTX-M+TEM also showed excessive numbers of isolates at site 2 and site 3. Please explain the reasons for the inconsistent data.

o The calculation of prevalence (%) was difficult to understand, e.g., the prevalence (%) of blaCTX-M from site 1 was calculated out of the total 125 isolates (96/125 = 76.8%), but the prevalence (%) of blaCTX-M-1 from site 1 was calculated out of 96 blaCTX-M-positive strains from site 1 (35/96 = 36.45). These could further lead to misinterpretation and misleading conclusions.

o Data from the last 3 rows was difficult to understand and was unlogic, for example, the number of blaCTX-M+TEM+SHV (the 3rd row from the bottom) at site 2 was 40, while the number of CTX-M (the 3rd row from the top) at site 2 was only 17.

Abstract section:

- Ln 11: “78.35% (n = 105)”: please make consistency with the Results section.

- Ln 8,9: “RT-PCR” was not a correct name of the method, and results of DNA-sequencing data was not reported in this manuscript.

- Ln 12-19: The prevalences of different genes were calculated among the ESBL-producing E. coli only (n = 125), not among the total isolated strains (n = 364). Additionally, the percentages of blaCTX-M-1 or blaCTX-M-15 were calculated out of the blaCTX-M-positive at each site, while the percentages of blaTEM and blaSHV were calculated out of 125 strains. Therefore, these sentences are arbitrary conclusions and are significantly misleading.

**Do you want your identity to be public for this peer review?** For information about this choice, including consent withdrawal, please see our Privacy Policy

Reviewer #1: **Yes: ** Abeni Beshiru

Reviewer #2: No

---

## [Author Response · Author response to Decision Letter 1]

27 Feb 2025

Dear Editor,

We would like to thank you for giving us the opportunity to revise our manuscript titled "Antimicrobial Resistance and Molecular Detection of Extended-Spectrum β-Lactamase (ESBL)-Producing Escherichia coli in Municipal Wastewater in Marrakech". We would like to thank the reviewers for their careful reading and thoughtful comments. We have carefully taken their comments, corrections, and suggestions into consideration in preparing our revision, which has resulted in a paper that is clearer and more compelling. All modifications are highlighted in the attached manuscript (text highlighted in yellow color).

Sincerely yours,

1.We note that your Data Availability Statement is currently as follows: [All relevant data are within the manuscript and its Supporting Information files.] Please confirm at this time whether or not your submission contains all raw data required to replicate the results of your study. Authors must share the “minimal data set” for their submission. PLOS defines the minimal data set to consist of the data required to replicate all study findings reported in the article, as well as related metadata and methods (https://journals.plos.org/plosone/s/data-availability#loc-minimal-data-set-definition).

Response 1: All relevant data are within the manuscript. We confirm that our submission contains all raw data required to replicate the results of our study, including the minimal data set .

Response 2 : We confirm that there are no ethical restrictions on sharing the data set used in this study, as no personal or sensitive information is included. Our study did not involve human subjects, and no ethics approval was required.

Reviewer(s)' Comments to Author :

Reviewer 1

The title should read “Antimicrobial Resistance and Molecular Detection of Extended-Spectrum β-Lactamase (ESBL)-Producing Escherichia coli in Municipal Wastewater in Marrakech: Implications for Public Health and Environmental Contamination.”

Response 1: We appreciate your valuable insight; however, we believe the original title maintains a concise and impactful representation of the study's main findings and objectives, which we feel resonates more effectively with the intended audience.

The abstract should be improved by incorporating the following: Revise sentences to avoid redundancy and enhance readability. For instance, the phrase: “emphasizing the persistence of these resistant genes even after disinfection processes” could be streamlined to: “highlighting the limited efficacy of disinfection in eliminating resistant genes.”Emphasize practical applications, e.g., how findings can inform wastewater treatment protocols. Avoid excessive numerical data in the abstract; instead, provide a general overview of key findings

Response 2: We recognize the suggestions provided. The abstract has been rewritten to enhance readability and conciseness. We have streamlined the language, reduced redundant numerical details, and added a sentence outlining the practical implications for wastewater treatment protocols.

The introduction could be improved as follows: Enhance the clarity of global relevance by briefly discussing similar findings in other regions.The literature review could benefit from integrating more recent studies (2023–2024) to reflect cutting-edge research on wastewater surveillance. State the research gap more explicitly, emphasizing why Morocco’s wastewater systems need targeted investigation.

Response 3 : We have added a brief discussion on recent studies (2023–2024) that report similar findings on ESBL-EC persistence in wastewater, highlighting global challenges in antimicrobial resistance (AMR) management. Additionally, we have explicitly addressed the research gap in Morocco.

The methods could be improved as follow : The rationale for selecting 72 samples could be expanded to address whether this number is statistically sufficient to generalize findings. While the description of tools and techniques (e.g., MALDI Biotyper, RT-PCR) is clear, specific details on reagent sources and controls need clarification to enhance reproducibility. Although the study involves wastewater samples, a brief mention of ethical considerations or permits for sample collection would strengthen compliance reporting. Include a section discussing data reliability and replication, especially for RT-PCR and sequencing steps. Clarify the storage or disposal methods for wastewater samples post-analysis

Response 4: The selection of 72 samples was based on a balance between feasibility, resource availability, and logistical constraints, ensuring both statistical power and practical execution of the study. Similar studies analyzing antibiotic resistance genes in wastewater have used comparable or smaller sample sizes while still drawing meaningful conclusions. (please see : doi: 10.1016/j.envint.2022.107171). A new subsection discussing data reliability, replication of PCR and sequencing, as well as the storage/disposal protocol for wastewater samples, has been added.

Tables 1 and Figures 1 and 2 contain excessive detailed data, which might be overwhelming for readers. Results focus heavily on resistance gene profiles but lack an in-depth interpretation of patterns and trends. Simplify tables by consolidating similar categories and highlighting the most relevant data. Include additional visual aids, such as heatmaps, to better illustrate resistance patterns across sample sites. Provide a brief summary for each result to connect data points to the research objectives.

Response 5: We sincerely appreciate your insightful suggestions. The tables and figures included in the manuscript present critical findings necessary for interpreting ESBL-producing E. coli prevalence, phenotypic characteristics, and resistance patterns. Each figure and table plays a distinct role in illustrating the results, and removing any of them could compromise the depth of the analysis. To maintain focus a summary statement has been added for each result section to connect the data to our study objectives. We have streamlined the formatting of Table 1 and ensured that key resistance gene distributions are summarized.

The discussion could be improved as follows: The section occasionally reiterates results instead of critically analyzing them. Lack of detailed suggestions on how findings could influence wastewater treatment improvements. Does not address the limitations of the study, e.g., the exclusion of seasonal variations or genetic typing limitations. Critically evaluate the persistence of genes post-disinfection and propose possible mechanisms.

Highlight practical interventions for improving wastewater treatment protocols based on findings.Discuss broader implications for global health surveillance, aligning with the One Health approach.Explicitly state study limitations and suggest future research directions, such as assessing resistance dynamics across seasons or expanding to other wastewater sites.

The conclusion could be improved as follows: Add a call to action for policymakers to tighten water treatment regulations. Suggest the adoption of advanced treatment technologies or routine monitoring programs. Highlight the need for interdisciplinary collaborations to address antibiotic resistance in wastewater.

Response 6: The discussion has been extensively revised to critically analyze the findings, acknowledge study limitations, and propose potential interventions. The conclusion includes a clear call to action for policymakers, recommendations for treatment technologies, and a discussion on the importance of interdisciplinary approaches to combat antibiotic resistance.

The Figures and tables could be improved as follows: Ensure figures are labeled with legends that clearly explain all components. Add captions that emphasize key takeaways for each table or figure. Reduce repetitive information to maintain reader engagement.The references could be improved as follows: Double-check for citation consistency, particularly with formatting styles. Include a few more references from 2023–2024 to strengthen the manuscript’s currency.

Response 7: All figures and tables have been revised with updated legends and captions to enhance clarity. We have also reviewed the reference list and incorporated several recent studies to strengthen the manuscript’s currency.

The manuscript grammar and sentence structures could be improved as follows

Revise long sentences for clarity and avoid excessive technical jargon where simpler language can suffice.Conduct thorough proofreading to eliminate typographical errors.

Response 8: The manuscript has been carefully proofread and all identified grammatical and typographical errors have been corrected.

Reviewer 2

Ln 92: please describe the effluent treatment method. In addition, the Results section mentioned about 2 stages of treatment: biological treatment and UV treatment, so please describe the details in this section.

Response 1: In response to your request, we have incorporated a detailed description of the effluent treatment method in the Sampling and Study Locations section to ensure clarity and consistency with the Results section.

Ln 101 : Please describe the “E. coli-like appearance” and cite the reference(s).

Response 2: Changes made. Please see the revised version. Quote ‘Colonies showing a flat, dry, pink, non-mucoid appearance with a darker pink zone of precipitated bile salts, typical of E. coli’.

Ln 102: please add the purpose of the MALDI method.

Response 3: We sincerely appreciate the reviewer's comment .I referenced established literature that thoroughly describes the purpose and application of the MALDI method. The primary function of MALDI-TOF MS in this study is to accurately identify Escherichia coli isolates, which is a well-documented and widely accepted application of this technique. Since the cited references provide detailed explanations of the method's purpose and mechanism, I aimed to maintain a concise narrative to avoid redundancy.

Ln 105: antimicrobial susceptibility test was performed on which strains, and on how many strains in total?

Response 4: AST was performed on ESBL-producing Escherichia coli isolates. We acknowledge the importance of reporting the total number of isolates; however, since the materials and methods section describes the experimental procedures, we believe that the appropriate place to report the final number of ESBL-EC isolates is in the Results section. In the Results section, we explicitly mention the total number of ESBL-producing E. coli isolates (n = 134) identified and subsequently tested for AST.

Ln 124–129 please add reference(s) for the Double Disk Synergy Test.

Response 5: Changes made. We have cited recent studies that describe the principles, methodology, and validation of DDST .

Ln 131: please note that “RT-PCR” stands for “Reverse transcription-PCR”, not “Real-time PCR”. Which method did the authors perform? Please also describe the PCR protocol after the boiling lysis step in this paragraph.

Response 6: Thank you very much for this well-pointed remarque .We have revised the text to explicitly state “Real-Time PCR” instead of "RT-PCR".We have now included the protocols following the boiling lysis step.

Ln 134: why did the authors choose the “susceptible” E. coli strain? How about the resistant strains?

Response 7: The susceptible E. coli strain was used in this study as a negative control to validate the specificity of qPCR amplification and to ensure that the primers did not produce non-specific amplification

Ln 139: what does “positive isolates” mean?

Response 8: The term "positive isolates" refers to E. coli isolates that tested positive for the presence of beta-lactamase genes (blaCTX-M, blaTEM, or blaSHV). To enhance readability, we revised the text to: Quote ‘Isolates that tested positive for beta-lactamase genes’

Ln 139–142: the authors mentioned about DNA sequencing and BlastN and BlastP analysis, but no data of these analyses were reported in the Results section!

Response 9: Thank you for your valuable feedback. In our study, we conducted a basic BLAST search to compare the obtained sequences with those in the NCBI and ARG-ANNOT libraries. Our objective was to verify the identity and similarity of the sequences rather than to discover novel sequences or mutations. In all cases, the obtained sequences showed 100% similarity to the corresponding sequences already present in these databases. Given this high level of similarity and the fact that no new or unique sequences were identified, we did not find it necessary to include detailed sequence data or submit the sequences.

Ln 149: please add the percentage (%) of resistant E. coli out of all resistant strains.

Response 10: We have added the percentage of resistant E. coli among all resistant isolates for clarity. As Total resistant isolates = 364 and Resistant E. coli isolates = 264 , This means 72.53% of all resistant isolates were resistant E. coli. The paragraph has been restructured for better readability while maintaining the logical flow of data.

Ln 150–151: typo mistakes, please add a word space before and after “;”

Response 11: Changes made. Thank you.

Ln 155 : please add the percentage (%) of ESBL-EC out of the resistant EC.

Response 12: We have calculated and incorporated the percentage in the text. Quote ‘ The phenotypic detection and the confirmation with selective Chrom ID ESBL resulted in the identification of 134 ESBL-EC (50.76%) among the 264 resistant E.coli. ’

Ln 155–159: was there any strain that was positive with Chrom ID ESBL but showed negative results with the Double Disk Synergy Test?

Response 13: We sincerely appreciate the reviewer's insightful comment .We did not observe any isolates that were positive on CHROMID® ESBL Agar but negative with DDST .

Ln 168–169: “Fifty percent of the isolated strains showed resistance to 3rd generation cephalosporins”, since the authors performed several steps with different methods in this study, please clarify the method and the number of “isolated strains” in this context.

Response 14: Thank you for your comment. In this context, the "isolated strains" refer specifically to the ESBL-producing E. coli (ESBL-EC) gathered after the phenotypic studies, including initial screening and confirmation using selective Chrom ID ESBL. Out of the 264 resistant E. coli strains, 134 (50.76%) were confirmed as ESBL-EC.

Ln 171–172: grammar error: “50% trimethoprim/sulfamethoxazole”. grammar error: “Meropenem at 1.56 % were less frequently observed”

Response 15: We have corrected two grammatical errors in the text

Ln 173: “78.40% (n = 98)” means that 100% were 125, but no population of 125 strains appeared from the previous results. So, from which population was 78.40% calculated? Meanwhile, the number written in the Abstract was 78.35% (n = 105). Please make consistency.

Response 16: The percentage of MDR-resistant bacteria was calculated based on a total population of 134 isolates used specifically for AST , out of which 105 were MDR-resistant. It is important to clarify that n=125 refers to the number of isolates analyzed for molecular analysis, which is a different population from the 134 isolates used for AST. This distinction has been clearly stated in the manuscript to avoid confusion.

Ln 175: “was covered”: did you mean “was discovered”?

Response 17: We changed the word to ‘was observed’.

Ln 187 and Table 1: the number of ESBL-producing E. coli was 134 but data in Table 1 showed n = 125. Please make a consistency.

Response 18 : A total of 134 ESBL-producing E. coli isolates were identified based on phenotypic confirmation using the Double Disk Synergy Test (DDST). However, only 125 of these isolates underwent molecular analysis. The difference (9 is

---

## [Decision Letter · Decision Letter 1]

8 Apr 2025

*Escherichia coli*

Dear Dr. Loqman,

Thank you for submitting your manuscript to PLOS ONE. After careful consideration, we feel that it has merit but does not fully meet PLOS ONE’s publication criteria as it currently stands. Therefore, we invite you to submit a revised version of the manuscript that addresses the points raised during the review process.

We look forward to receiving your revised manuscript.

Kind regards,

Gabriel Trueba, PhD

Academic Editor

PLOS ONE

Reviewers' comments:

Reviewer's Responses to Questions

**Comments to the Author**

Reviewer #2: All comments have been addressed

Reviewer #3: All comments have been addressed

Reviewer #4: All comments have been addressed

2. Is the manuscript technically sound, and do the data support the conclusions?

Reviewer #2: Yes

Reviewer #3: Yes

Reviewer #4: Partly

3. Has the statistical analysis been performed appropriately and rigorously?

Reviewer #2: Yes

Reviewer #3: Yes

Reviewer #4: I Don't Know

4. Have the authors made all data underlying the findings in their manuscript fully available?

Reviewer #2: Yes

Reviewer #3: Yes

Reviewer #4: No

5. Is the manuscript presented in an intelligible fashion and written in standard English?

Reviewer #2: Yes

Reviewer #3: Yes

Reviewer #4: No

Reviewer #2: (No Response)

Reviewer #3: On account of the manuscript PONE-D-24-59416R1, entitled “Antimicrobial Resistance and Molecular Detection of Extended-spectrum ß-Lactamase (ESBL)-Producing Escherichia coli in Municipal Wastewater in Marrakech” by Oumaima EL GARRAOUI et al., the authors revised the manuscript intensively and appropriately according to the Reviewers comments. After careful consideration, I made a decision that the manuscript is acceptable for publication in its present form.

Reviewer #4: The authors have attempted to answer the reviewer's comment; however, there is still room for improvement as far as the manuscript is concerned.

kindly find below comments

Line 10: How many samples resulted in 364 resistant Enterobacteriaceae

Line 12; “in influent samples” what do you mean by influent samples? These genes are found in the bacteria and are not necessary in the sample. kindly clarify.

Line 13: “Although biological treatment reduced resistant gene abundance.” This statement is not necessarily true as it should refer to the bacteria and not the genes.

Lines 39-31: The meaning of that paragraph is not clear. Kindly change the word “However” to “As” to make the statement a little clearer.

Line 81; “Colonies showing a flat, dry, pink…” From which MacConkey agar plate? The one with or the one without 4 mg/L of ceftriaxone. Kindly make it easy for readers to follow.

Line 101; “AST results were further confirmed using the Phoenix™ Automated Microbiology System (Becton–Dickinson Diagnostic Systems, Sparks, MD, USA” Why was the AST confirmed again?

Line 111-112: An interactive result between clavulanic acid and one of the chosen cephalosporins on growth was identified as positive for ESBL production.’ What does that mean; interactive result

Line 120- check the reference style

Line 143-146: “A total of 72 wastewater samples were collected from influent (n = 25) and treated effluent (n = 47 this section is not clear; how many samples did you take. (25+47=72) . Kindly put into the bracket (), including biologically treated effluent or secondary (n = 24) and tertiary treatment effluent (n = 23).

Line 147- ??

Line 151: “This indicates that wastewater treatment, particularly tertiary treatment, effectively reduces the proportion of resistant isolates but does not eliminate them completely.” this sounds like discussion, not result

Line 153 “The phenotypic detection and the confirmation with selective Chrom ID ESBL resulted in the” why was the Chrom ID used after you have ID with MaldiTOf is there a discrepancy how was it solve.

Line 167 “Fifty percent” This should be numerical

Your table is a bit unclear, I am struggling to understand prevalence here as opposed to frequency. Do you mean frequency among your samples/isolates or prevalence/There is a clear distinction between these two words, and I will be glad if you can make them clear here?

Line 191-206; is more of discussions than results

Line 227; ” Although chlorine disinfection and UV treatment decrease bacterial loads, research indicates” where was this shown in the result section.

**Do you want your identity to be public for this peer review?** For information about this choice, including consent withdrawal, please see our Privacy Policy

Reviewer #2: **Yes: ** Mi Nguyen-Tra Le

Reviewer #3: No

Reviewer #4: **Yes: ** Dr. Charity Wiafe Akenten

---

## [Author Response · Author response to Decision Letter 2]

28 Apr 2025

Response Letter

Manuscript ID : PONE-D-24-59416R1

EMID : 224006d43d0c6d21

Dear Editor,

We would like to express our sincere gratitude for the opportunity to revise our manuscript entitled "Antimicrobial Resistance and Molecular Detection of Extended-Spectrum β-Lactamase (ESBL)-Producing Escherichia coli in Municipal Wastewater in Marrakech." We also extend our thanks to the reviewers for their thoughtful and constructive comments, as well as their valuable suggestions. We have carefully addressed all of the reviewers’ comments in this revised version and believe that the changes have substantially improved the clarity and overall quality of the manuscript. All modifications are clearly marked in the revised document, with the updated text highlighted in yellow for easy reference.

Sincerely yours,

Reviewer(s)' Comments to Author :

Reviewer 4

Line 10: How many samples resulted in 364 resistant Enterobacteriaceae ?

Response 1: Thank you for your comment. We have clarified this point in the revised manuscript. The updated text now reads : “From a total of 72 wastewater samples, 364 resistant Enterobacteriaceae isolates were recovered, of which 134 were confirmed as ESBL-producing E. coli.”

Line 12 : “in influent samples” what do you mean by influent samples? These genes are found in the bacteria and are not necessary in the sample. kindly clarify.

Response 2: We appreciate your observation. We have revised the sentence to clarify that the resistance genes were detected in bacterial isolates obtained from influent samples. By “influent samples,” we refer to raw wastewater collected at the inlet of the treatment plant, prior to undergoing any treatment processes.

Line 13: “Although biological treatment reduced resistant gene abundance.” This statement is not necessarily true as it should refer to the bacteria and not the genes.

Response 3: Thank you for your valuable observation. Our reference to a reduction in "resistant gene abundance" was intended to indicate a decrease in the number of bacterial isolates carrying these genes. To improve clarity and accuracy, we have revised the sentence to read: “Although biological treatment reduced the number of ESBL-producing E. coli isolates harboring resistance genes.”

Lines 39-31: The meaning of that paragraph is not clear. Kindly change the word “However” to “As” to make the statement a little clearer.

Response 4 : Thank you for your suggestion. We have revised the sentence to enhance clarity, as recommended. The updated text now reads: “As wastewater treatment plants (WWTPs) often fail to achieve complete microbial removal, antibiotic-resistant bacteria (ARB) and antibiotic resistance genes (ARGs) may persist in treated effluents.”

Line 81 :“Colonies showing a flat, dry, pink…” From which MacConkey agar plate? The one with or the one without 4 mg/L of ceftriaxone. Kindly make it easy for readers to follow.

Response 5 : We sincerely appreciate the reviewer’s comment. We have clarified in the revised manuscript that the described colonies were selected from MacConkey agar plates supplemented with 4 mg/L of ceftriaxone, as these were used specifically to isolate ceftriaxone-resistant E. coli.

Line 101: “AST results were further confirmed using the Phoenix™ Automated Microbiology System (Becton–Dickinson Diagnostic Systems, Sparks, MD, USA” Why was the AST confirmed again?

Response 6: Thank you for your question. In our laboratory, the Phoenix™ Automated Microbiology System is routinely used alongside the disk diffusion method to confirm antimicrobial susceptibility profiles. This dual approach enhances the reliability and reproducibility of results, especially when detecting multidrug-resistant organisms such as ESBL-producing E. coli. Therefore, both methods were included in the study to ensure data robustness and quality assurance.

Line 111-112 : An interactive result between clavulanic acid and one of the chosen cephalosporins on growth was identified as positive for ESBL production.’ What does that mean; interactive result

Response 7 : Thank you for your valuable comment. In this context, “interactive result” refers to the synergistic effect observed during the Double-Disk Synergy Test (DDST). Specifically, a keyhole-shaped enhancement or an increase in the inhibition zone between the clavulanic acid disk and any of the cephalosporin disks (cefotaxime, ceftazidime, or cefepime) was interpreted as a positive result for ESBL production. We have revised the sentence accordingly to improve clarity.

Line 120 : check the reference style

Response 8: Thank you for pointing this out. The reference style has been corrected according to the journal's guidelines.

Line 143-146 : “A total of 72 wastewater samples were collected from influent (n = 25) and treated effluent (n = 47 this section is not clear; how many samples did you take. (25+47=72) . Kindly put into the bracket (), including biologically treated effluent or secondary (n = 24) and tertiary treatment effluent (n = 23).

Response 9 : Thank you for your helpful comment. We have revised the sentence for clarity. The updated version now reads: “A total of 72 wastewater samples were collected, including influent (n = 25), biologically treated effluent or secondary treatment (n = 24), and tertiary treatment effluent (n = 23).”

Line 147- ??

Response 10 : Thank you for your comment. We have clarified this section in the revised manuscript. The 364 resistant Enterobacteriaceae isolates were recovered from MacConkey agar supplemented with 4 mg/L ceftriaxone. Among these, 264 isolates were identified as Escherichia coli using the MALDI-TOF Biotyper® system and were considered resistant based on their growth on selective media. The revised sentence now reads: “Out of 364 resistant Enterobacteriaceae isolates recovered from selective agar, 264 were confirmed as resistant E. coli, representing 72.52% of the total resistant isolates.”

Line 151: “This indicates that wastewater treatment, particularly tertiary treatment, effectively reduces the proportion of resistant isolates but does not eliminate them completely.” this sounds like discussion, not result

Response 11 : Thank you for your comment. We have revised this sentence to better align with the results section. The updated version now focuses on the observed findings.

Line 153 :“The phenotypic detection and the confirmation with selective Chrom ID ESBL resulted in the” why was the Chrom ID used after you have ID with MaldiTOf is there a discrepancy how was it solve.

Response 12 : Thank you for your insightful comment. There is no methodological discrepancy. MALDI-TOF was employed for species identification of Escherichia coli, while CHROMID® ESBL agar was used for phenotypic confirmation of ESBL production among the E. coli isolates. This sequential approach is a standard practice, ensuring both accurate species identification and reliable resistance profiling.

Line 167 :“Fifty percent” This should be numerical .

Your table is a bit unclear, I am struggling to understand prevalence here as opposed to frequency. Do you mean frequency among your samples/isolates or prevalence/There is a clear distinction between these two words, and I will be glad if you can make them clear here.

Response 13 : We appreciate your feedback. The values presented in the table reflect the frequency of resistance genes among confirmed ESBL-producing E. coli isolates at each site. To enhance clarity and accuracy, we have replaced the term “prevalence (%)” with “frequency (%)” in the table header and updated the corresponding caption accordingly.

Line 191-206 : is more of discussions than results

Response 14 : Thank you for your observation. In response, we have revised the text to focus solely on the observed findings, ensuring it is more appropriate for the Results section.

Line 227 :” Although chlorine disinfection and UV treatment decrease bacterial loads, research indicates” where was this shown in the result section.

Response 15 : Thank you for your comment. We have revised the sentence to specifically refer to resistant isolates, in line with the findings presented in the Results section. The updated version now reads: “Although chlorine disinfection and UV treatment decrease the number of resistant E. coli isolates…”

---

## [Decision Letter · Decision Letter 2]

20 May 2025

Antimicrobial Resistance and Molecular Detection of Extended-spectrum ß-Lactamase (ESBL)-Producing *Escherichia coli* in Municipal Wastewater in Marrakech

PONE-D-24-59416R2

Dear Dr. Loqman,

We’re pleased to inform you that your manuscript has been judged scientifically suitable for publication and will be formally accepted for publication once it meets all outstanding technical requirements.

Kind regards,

Gabriel Trueba, PhD

Academic Editor

PLOS ONE

Additional Editor Comments (optional):

Reviewers' comments:

Reviewer's Responses to Questions

**Comments to the Author**

Reviewer #4: All comments have been addressed

2. Is the manuscript technically sound, and do the data support the conclusions?

Reviewer #4: Yes

3. Has the statistical analysis been performed appropriately and rigorously?

Reviewer #4: Yes

4. Have the authors made all data underlying the findings in their manuscript fully available?

Reviewer #4: Yes

5. Is the manuscript presented in an intelligible fashion and written in standard English?

Reviewer #4: Yes

Reviewer #4: The authors have satisfactorily addressed all the reviewers’ comments and questions. The revisions have improved the clarity and quality of the manuscript. I have no further concerns.

**Do you want your identity to be public for this peer review?** For information about this choice, including consent withdrawal, please see our Privacy Policy

Reviewer #4: **Yes: ** Charity Wiafe Akenten

---

## [Editor Report · Acceptance letter]

PONE-D-24-59416R2

PLOS ONE

Dear Dr. Loqman,

I'm pleased to inform you that your manuscript has been deemed suitable for publication in PLOS ONE. Congratulations! Your manuscript is now being handed over to our production team.

Kind regards,

on behalf of

Dr. Gabriel Trueba

Academic Editor

PLOS ONE